# Linked surveillance and genetic data uncovers programmatically relevant geographic scale of Guinea worm transmission in Chad

**Jessica V. Ribado**[1], **Nancy J. Li**[1], **Elizabeth Thiele**[2], **Hil Lyons**[1], **James A. Cotton**[3], **Adam Weiss**[4], **Philippe Tchindebet Ouakou**[5], **Tchonfienet Moundai**[5], **Hubert Zirimwabagabo**[4], **Sarah Anne J. Guagliardo**[4,6], **Guillaume Chabot-Couture**[1], **Joshua L. Proctor**[1] *

**1** Institute for Disease Modeling, Global Health Division of the Bill and Melinda Gates Foundation, Seattle, Washington, United States of America, **2** Vassar College, Poughkeepsie, New York, United States of America, **3** Wellcome Sanger Institute, Hinxton, Cambridgeshire, United Kingdom, **4** The Carter Center, Atlanta, Georgia, United States of America, **5** National Guinea Worm Eradication Program, Ministry of Public Health, N'Djamena, Chad, **6** Centers for Disease Control and Prevention, Atlanta, Georgia, United States of America

* jproctor@idmod.org

**Data Availability Statement:** Genomic sequences are available on GenBank as MW257241 - MW257699 for CO3, MW257700 - MW258158 for

## Abstract

### Background

Guinea worm (*Dracunculus medinensis*) was detected in Chad in 2010 after a supposed ten-year absence, posing a challenge to the global eradication effort. Initiation of a village-based surveillance system in 2012 revealed a substantial number of dogs infected with Guinea worm, raising questions about paratenic hosts and cross-species transmission.

### Methodology/principal findings

We coupled genomic and surveillance case data from 2012-2018 to investigate the modes of transmission between dog and human hosts and the geographic connectivity of worms. Eighty-six variants across four genes in the mitochondrial genome identified 41 genetically distinct worm genotypes. Spatiotemporal modeling revealed worms with the same genotype ('genetically identical') were within a median range of 18.6 kilometers of each other, but largely within approximately 50 kilometers. Genetically identical worms varied in their degree of spatial clustering, suggesting there may be different factors that favor or constrain transmission. Each worm was surrounded by five to ten genetically distinct worms within a 50 kilometer radius. As expected, we observed a change in the genetic similarity distribution between pairs of worms using variants across the complete mitochondrial genome in an independent population.

### Conclusions/significance

In the largest study linking genetic and surveillance data to date of Guinea worm cases in Chad, we show genetic identity and modeling can facilitate the understanding of local transmission. The co-occurrence of genetically non-identical worms in quantitatively identified

cytB, and MW258159 - MW258617 for ND3-5 genes. Analysis code can be accessed at https://github.com/InstituteforDiseaseModeling/GWSpatialGenetics. The epidemiological data containing the case coordinates must be requested through The Carter Center for approval by the Chad Ministry of Public Health. For epidemiological data requests, contact The Carter Center (info@cartercenter.org) or Yujing Zhao, Data Analyst for the Guinea Worm Eradication Program at The Carter Center (yujing.zhao@cartercenter.org).

**Funding:** This publication is based on research funded in part by the Bill & Melinda Gates Foundation, including but not limited to models and data analysis performed by the Institute for Disease Modeling at the Bill & Melinda Gates Foundation by JVR, NL, HL, GCC, JLP. JAC was supported by funding from The Carter Center and Wellcome, via their core support for the Wellcome Sanger Institute (grant WT206194). The funders had no role in study design, data collection and analysis, decision to publish, or preparation of the manuscript.

**Competing interests:** The authors have declared that no competing interests exist.

transmission ranges highlights the necessity for genomic tools to link cases. The improved discrimination between pairs of worms from variants identified across the complete mitochondrial genome suggests that expanding the number of genomic markers could link cases at a finer scale. These results suggest that scaling up genomic surveillance for Guinea worm may provide additional value for programmatic decision-making critical for monitoring cases and intervention efficacy to achieve elimination.

## Author summary

The global eradication effort for Guinea worm disease has dramatically decreased the global burden of the disease and enabled 187 countries to be certified by the World Health Organization to be free of endemic transmission. Despite this progress, several countries continue to have endemic transmission. In Chad, a long absence of reported cases was interrupted with the identification of new Guinea worm cases, prompting a substantial scale up of surveillance and intervention efforts. Here, we study the value of increasing genomic surveillance as a tool for programmatic evaluation of surveillance and intervention efforts in Chad. Linking surveillance and genomic samples, parsimonious spatial models help reveal a consistent geographic clustering of similar genetic sequences across Chad. We also demonstrate that expanding the sequencing can offer better resolution for distinguishing Guinea worm samples. In this retrospective study, we found evidence that scaling up genomic surveillance can be an important monitoring and evaluation tool for the eradication program in Chad.

## Introduction

The eradication campaign for dracunculiasis, Guinea worm disease, has made substantial progress since the first set of World Health Assembly resolutions aimed at elimination efforts passed during the International Drinking Water Supply and Sanitation Decade (1981–1990) [1]. The Guinea worm eradication campaign has decreased the global burden of disease by more than 99% [2], enabled the World Health Organization (WHO) to certify 187 countries as free from endemic dracunculiasis transmission [3], and decreased the significant economic loss associated with dracunculiasis in rural settings [1]. Despite the substantial progress, endemic transmission has persisted in Angola, Chad, Ethiopia, Mali, and South Sudan [2]. In Chad, a decade-long absence of reported cases was interrupted with a detected resurgence of Guinea worm cases in 2010 [4]; human cases continue to be reported [5] with a growing understanding of the role of animal reservoirs in transmission [6–10]. The stalled progress in Chad has led to a substantial increase in surveillance and programmatic efforts [11] as well as investments in research and novel interventions [12, 13].

Eradication and control for a variety of pathogens rely on genetic analysis tools. In the case of polio eradication, for example, these tools have been used to detect silent transmission across geographic areas and thus inform programmatic decision-making [14–16]. The inclusion of strain differentiation techniques into programmatic decision-making for poliomyletis is feasible due to the continual mutation events in the virus genome, a growing global library of samples and isolates, and a mature set of mathematical and statistical methodologies to track specific lineages. Well-established phylogenetic methodologies to infer high-fidelity ancestry and lineages do not yet exist for parasites such as Guinea worm [10, 17]. Nonetheless,

genetic data collected from Guinea worms in Chad has already revealed research and programmatically-relevant insights: human and dog hosts share a common genetic population suggesting transmission between species [10]. Genome-wide data from a much smaller sample of worms confirms this finding [12]. We build upon those research insights using linked genetic and surveillance data from Chad to investigate the geographic connectivity of genetically defined worm populations. Spatial epidemiological models have been a key tool for inferring the connectivity of different populations and providing insights into the propagation of infectious diseases. Spatial models, such as the gravity [18] and Levý flight [19], have been utilized to describe the movement of humans and animals as a key component to the spread of infectious diseases [20–26]. These models and sophisticated inference algorithms have helped characterize the movement of pathogens and provide insights for programmatic decision-making [27, 28]. Genetic data can be used to indirectly study the movement of pathogens, bypassing the need for host movement data [16, 29–31]. Using genetic and epidemiological data to inform infectious disease models is a growing field, which encompasses phylodynamics [32–35]. Similar to phylogenetic methodologies, phylodynamic analyses and models are most mature for virus and bacteria pathogens [16, 27, 30, 31, 36]. Recent innovations of using the biological characteristics of parasites clonally propagating in order to build phylodynamic models have been applied for malaria parasite movement within neighborhoods of Thiés, Senegal [37]. We demonstrate that probabilistic spatial models informed by Guinea worm genetic data can reveal new insights into the geographic connectivity of worm populations in Chad.

In this article, we investigate the programmatic potential of leveraging genetic data for enhanced surveillance efforts by performing a retrospective analysis and modeling of the epidemiological and genomic data collected in Chad from 2012–2018, excluding 2014. We leverage both previously reported [10] and new genetic sequences linked with surveillance data to build spatial models that reveal a consistent geographic connectedness of worm populations in Chad. We also show expanding sequencing across the mitochondrial genome for Guinea worm changes population pairwise similarity, which could provide higher resolution in linkages between samples. These results are followed by a discussion on the implications of increasing the scope of genetic sequencing for programmatic surveillance and decision-making.

## Materials and methods

### 0.1 Ethics statement

The World Health Organization and the Chad Ministry of Public Health (MOPH) have sanctioned the extraction of emergent adult worms to interrupt the life cycle and prevent environmental contamination with larvae. Trained program and ministry staff extracted female worms from active cases as routine public health protocol to contain and treat Guinea worm infections. The extracted worm is a by-product of standard disease treatment and intervention. Usage of the material for genetic analysis is sanctioned by the country MOPH. Worms were stored in ethanol as described in Eberhard et al. [38]. Samples from human cases are anonymized prior to study inclusion.

### 0.2 DNA extraction and sequencing

Extraction was attempted on 712 worms collected in Chad from 2012–2018, excluding samples from 2014 due to availability. Whole genomic DNA was extracted from 5–15mm sections of adult female worm tissue with one of two methods: a modified Puregene DNA extraction protocol as detailed in Thiele et al. 2018 [10], or with the DNeasy Blood and Tissue Kit (Qiagen, Germantown, MD, #69582) according to the manufacturer's protocol for tissue extraction.

Five hundred and ninety-five samples were successfully sequenced at one or more of the four targeted mitochondrial genes (CO3, cytB, and ND3–5). Sanger sequencing and base calls from chromatograms were performed as detailed in Thiele et al. 2018 [10]. Four hundred and sixty-one samples were successfully sequenced at all four targeted genes. Thirty three of these samples were first reported in a previous assessment of genetic diversity of Chadian Guinea worms (S1 Table) [10]. The remaining 428 samples have not been published previously.

### 0.3 Alignment and variant identification

**0.3.1 Targeted CO3, cytB, ND3–5 genes.**   Sequences for CO3, cytB, and ND3–5 for 461 worms were aligned to the *Dracunculus medinensis* mitochondria genome version JN555591.1 from the European Nucleotide Archive with the BWA v0.7.17 software package [39] to confirm amplification of the correct genes. Amplification fragments spanned the genomic coordinates of 3, 690 to 4, 308 for CO3, 2, 628 to 3, 234 for cytB, and 12, 562 to 14, 523 for ND3–5. Bases (A, C, G, T, or other for ambiguous or missing) were counted for each position in the alignment ranges for the population. Missing bases were concentrated at the beginning and/or ends of the gene ranges due to amplification variation. Positions in the alignment ranges were excluded from variant discovery if more than two samples with successful amplification for that gene had the designation of missing or ambiguous.

Positions with at least two different bases (excluding ambiguous) were variant candidates and checked for irregularities. Two worm samples exhibited highly irregular singleton characteristics, contributing more than 50% of the singleton variants in a single gene. These two worm samples were excluded from variant discovery and analyses. From the remaining worms, we identified 86 variant positions concatenated to create a molecular barcode. Forty-one unique barcodes were identified across 459 worms. Six variants in the barcode were singletons in the population. Three barcodes contained an ambiguous base 'N', assigned to four samples in the population. Barcodes were assigned an identifying number based on the number of samples belonging to each barcode. For methods describing the accumulation of barcode diversity in the population at different genes, refer to S1 Appendix.

**0.3.2 Complete mitochondria genome.**   Next-generation shotgun sequencing from nineteen publicly available *Dracunculus medinensis* mitochondrial genomes were downloaded from the European Nucleotide Archive (PRJEB1236) [12]. Low quality bases with a minimum mapping quality of 20 were trimmed from the ends of reads. Trimmed reads were aligned to the *D. medinensis* mitochondria genome with the BWA v0.7.17 software package [39]. Variants were called on de-duplicated, uniquely aligned reads following best practices outlined by GATK v4.1.4. Known variants are typically recommended to correct sequencing errors that lead to spurious variant calls. A set of known variants is not available for the *D. medinensis* mitochondria genome. Instead, bootstrap base recalibration was done using higher confidence calls (QD $>$2.0, FS $>$60.0, MQ $<$40.0, MQRankSum $<-$12.5, ReadPosRankSum $<-$8.0) until the final calls converged (2–3 steps) [40]. Final variants were filtered using the following input parameters for GATK: FS $<=$ 13.0 or missing, SOR $<$3.0 or missing, and $-$3.1 $<=$ ReadRankPos, BaseQRankSum, MQRankSum, ClippingRankSum $<=$ 3.1 [12]. Given the low sample size for recalibration, orthogonal variant calling was performed using the bcftools program [41]. Genotype maximum likelihoods were determined using the mpileup command with parameters specifying a minimum mapping quality of 20 and ploidy of 1. For both variant call sets, variants found within positions 5950–6670 were removed due to high heterozygous calls likely from heteroplasmy and 2× mapping coverage. For all other variants, a minimum depth of 10 reads was required across all samples to be included in downstream analyses.

## 0.4 Clustering of molecular barcodes

Discriminant analysis of principal components (DAPC) was used to infer population structure based on the molecular barcodes using the R package adegenet v2.1.3 [42–44]. Populations were defined by host identity. Barcodes found in samples from humans only were grouped with barcodes found in samples from humans and dogs to create more even groups (humans only = 5 barcodes, both = 12, dogs only = 24). To investigate whether the barcodes differentiated by host species, we specified one discriminant axis for the two classifications. We investigated the optimal number of principal components based on analyzing the distribution of eigenvalue magnitudes and leveraging cross-validation to evaluate the lowest root mean square error (RMSE) for discrimination. Due to the small number of unique barcodes, we tested the robustness of the RMSE estimates using 70%, 80%, and 90% of the data as the training set with 1000 permutations each. We observed the xvalDapc function default of a 90% training set gave the lowest RMSE estimate.

## 0.5 Linking genomic samples to epidemological data

We linked the sequenced samples with corresponding surveillance data using national case reports and standardized surveys, described in detail in [45]. Case data was collected from an active and passive village-based surveillance system by the Chad Guinea Worm Eradication Program (CGWEP) and the Ministry of Public Health [46]. Summary case reports at the village level were compiled across surveillance sites by the CGWEP program. Information on infected dogs was collected through a standardized survey with the self-identified owner. Where available, global position system (GPS) coordinates from the standardized surveys were validated with village coordinates from the national case reports. The GPS coordinates were consistent across national case reports and standardized surveys for 207 samples. In instances where coordinates differed between the standardized surveys and the national case report, the national case report coordinates were assigned (181 samples). For cases reported with a village name in the standardized survey but without GPS coordinates, the GPS coordinates associated with the village name in the national case report was assigned to that sample (2 samples). For cases reported with a village name and GPS coordinates, but the village name was not in the national case report, standardized survey case coordinates were assigned to that sample (37 samples). Four hundred and twenty-six worms were linked across 245 hosts.

## 0.6 Probabilistic spatial models of Guinea worm connectivity

**0.6.1 Spatial models.** Spatial movement models have demonstrated that distance is an important factor influencing the spread of pathogens. [21–23, 30]. Here, due to the yearly life cycle of the Guinea worm parasite [1], we specified a spatial model similar to a probabilistic diffusion model for a discrete spatial network representing the villages affected by Guinea worm in Chad. The model describes the probability of a worm being transmitted from village $i$ to village $j$ based on the geographic distance $d$ between a pair of worms such that $p_{ij} = C_i f(d_{ij})$. $C_i$ is a scaling factor and f is informed by the pairwise geographic distance between pairs of worms with identical or non-identical barcode identity. Similar to other parametric formulations such as the diffusion, gravity [22], or Lévy flight models [30], our non-parametric formulation could be used to predict where a future linked Guinea worm case might appear.

**0.6.2 Genetic identity and geographic distance between samples.** To infer f in the model above, we used the genetic identity between samples and the associated GPS coordinates for villages. We estimated the number of variant differences across the mitochondrial genome that may connect a direct lineage (i.e. a maternal-offspring pair of worms) from a spontaneous mutation rate. Without a known organismal mutation rate, we assumed a similar mutation

rate as the *Caenorhabditis elegans* spontaneous mitochondrial mutation rate estimated at 1.05$e$−7 site/generation [47]. From this rate, we expected 0.0015 mutations for the 14,628 base pair *Dracunculus medinensis* mitochondrial genome between generations. Given this extrapolated generational mutation rate of less than one, worms were grouped by exact barcode identity. We did not make any assumptions of pairwise genetic distance from models of DNA evolution for analyses comparing genetic similarity between pairs of worms. We calculated the pairwise genetic similarity as 1—the Hamming distance from the molecular barcodes to obtain the number of shared variants [48]. Haversine distance was used to compute the geographic distance between each pair of worms $d$ using the R package geosphere v1.5 [49].

Grouping barcodes by exact identity may exclude pairs of samples that share a recent lineage in the event of an underestimated mutation rate or an unlikely, but possible, sequencing error. We tested the impact of a single base pair difference in barcodes between a pair of worms on geographic clustering (see section 0.6.4). Relaxing thresholds on a pairwise basis avoided issues clustering barcodes with more than a single base pair difference. For example, barcodes A and B had two base pair differences, but barcode A and C and barcode B and C each have a single base pair difference.

**0.6.3 Spatial models by molecular barcode sets.**   To characterize f, we partition the samples according to their genetic identity. Samples with identical barcodes are grouped; for each barcode $k$, all of the pairwise geographic distances within the set are used to compute a barcode functional form $f_k$. The set of functions $f_k$ are then averaged to produce a single f. We also investigated the impact of partitioning samples with a less restrictive criterion based on the assumed mutation rate as described above. Similar to research in the comparison of power-law distributions [50], we utilized the empirical cumulative densities f using stat_ecdf and compared distributions using the Kolmogorov-Smirnov test with base R v3.6.3 [51]. We plotted the empirical distributions as smoothed kernel density distributions using the function stat_density contained in the R package ggplot2 v3.3.0 [52].

**0.6.4 Sensitivity analyses.**   We investigated the results with sensitivity analyses including statistical bootstrapping. Distance permutations between pairs of worms included 100 iterations of sampling without replacement for the all-by-all pairwise geographic distances. Multi-infected host bootstrap subsampling included 100 iterations of choosing a single case worm at random for each host. In instances where only one case is observed per host, that case was consistently the representative. Each iteration was considered an independent distribution and analyzed as described above. In addition, we also tested the impact of relaxed similarity thresholds for defining matched identity of the molecular barcodes between pairs on the geographic clustering.

**0.6.5 Analyzing the diversity of genetic samples by geography.**   A cumulative barcode diversity score was calculated for each worm with GPS coordinate data (426 worms). The number of unique barcodes was counted for an expanding radius around each worm until every other worm in the study was included in the radius. We also investigated the effect of opportunistic sampling on barcode geographic clustering by comparing the cumulative distance from each case to every other worm in the population. From each worm, we expected the number of samples to increase at variable rates depending on sample clusters. Deviations from an increase (i.e. a sustained flat line) between samples would indicate geographic breaks between sample clusters.

## 0.7 Analyzing sensitivity of additional regions outside of CO3, cytB, and ND3–5

Variants in the complete mitochondrial genome were grouped by whether the variant was in the CO3, cytB, and ND3–5 gene ranges ('targeted genes') or in other regions of the genome

('untargeted genes', including non-coding regions). Gene ranges were determined by the start and end positions of genes in the *Dracunculus medinensis* mitochondria genome version JN555591.1 instead of the amplicon alignment ranges due to the differences in sample processing and sequencing technology. The genomic ranges for the genes are 3,778–4,543 for CO3, 2,619–3,720 for cytB, and 12,550–14,467 for ND3–5. Pairwise genetic similarity was calculated from barcodes created with all variants, variants within targeted gene ranges, or variants within untargeted gene ranges. For bootstrap subsampling tests, variants within untargeted genes were randomly selected to contain the same number of variants found within targeted genes to create a new barcode for genetic similarity comparisons. Pairwise genetic similarity comparisons for the different gene groups were robustly tested across two variant calling pipelines (GATK and bcftools mpileup, see 0.3.2) with all variants and singleton variants excluded. Smooth kernel density distributions were generated with ggplot2 v3.3.0 option "stat_density". A two sided Kolmogorov-Smirnov test was used to compare genetic similarity distributions of different gene groups with base R v3.6.3 [51].

# Results

## 0.8 Genetic and epidemiological characteristics of Guinea worm cases

Four hundred and fifty-nine worms (30 worms from humans hosts, 429 worms from dog hosts) collected from 2012–2018 were successfully sequenced (Fig 1A). Four hundred and twenty-six successfully sequenced worms were matched to GPS coordinates for each reported case. Worm cases mainly clustered along the Chari River (Fig 1A). Identical mitochondrial sequences were observed for cases across years (Fig 1B). Eighty six variants were identified in the CO3, cytB, and ND3–5 genes of 618, 606, and 1,961 base pairs, respectively. The concatenation of the 86 variants resulted in 41 unique molecular barcodes (S2 Table).

Barcode accumulation curves suggest the 41 barcodes in this study did not saturate the potential diversity of the unobserved population. We estimated the number of barcodes that may be uncovered with more sequencing. A standard negative binomial and an empirical Bayes approach [53] estimated we may observe 25–40 additional barcodes for a sampled population size of 5000. Despite the differences between methods, both indicated that modestly increasing the number of samples sequenced is not likely to capture the complete underlying genetic diversity. However, the extrapolation was hindered by the available data; it is unclear whether the abundance of low-frequency barcodes in the currently sampled population are representative of the population. See S1 Appendix for further methodological details and numerical investigations.

## 0.9 Molecular barcode similarity and persistence between host species

Twelve of the 41 identified barcodes were shared between human and dog hosts. There were more unique dog barcodes than human barcodes driven by the high number of reported cases in dogs (Fig 1A). From the years included in this study, it does not appear worms were shifting from one host species to the other (S1(A) Fig). Barcodes were likely to appear in dogs and humans in the same year (barcodes 2, 6, 8, 10, 36), dogs a year prior to humans (barcodes 5, 7, 13, 21), or in humans a year prior to dogs (barcodes 3, 4, 14).

In addition, DAPC of barcodes highlighted a challenge to discriminate between lineages of human and dog hosts. The rate of cumulative variance decreases drastically after the first four principal components (S1(B) Fig). Cross-validation identified 15 components with the lowest RMSE across different training set ranges (0.44, S1(C) Fig). The overlap of discriminant distributions predicted with five or fifteen principal components suggests molecular barcodes do

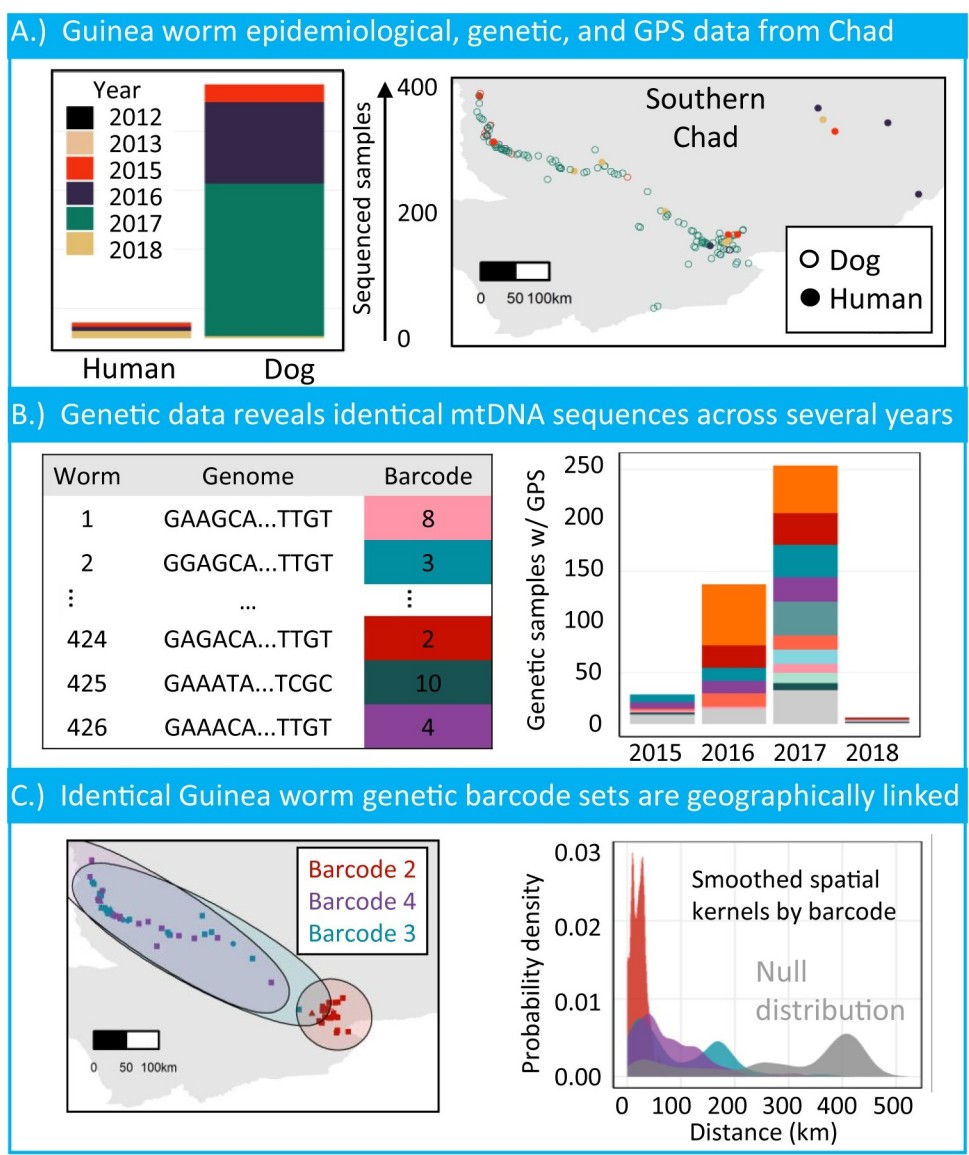

**Fig 1. Characteristics of genetically characterized Guinea worm cases in Chad from 2012–2018.** A.) Number of cases per host species collected from 2012–2018, with GPS matched sample distributions across the Chari River. Chad maps were generated with GADM data. B.) The number of GPS matched samples that belong to barcode sets. Note, not all barcode colors are shown in the left figure. Barcodes with less than 10 samples in the population are colored in light gray for visual clarity. C.) Samples belonging to barcode sets 2, 3 and 4 are shown in their respective ranges. Smoothed kernel densities for the three barcode sets show the mass of the distributions correlate with the spatial connectedness of samples.

not differ by host (S1(D) Fig). The mean successful assignment was 48% and 61% respectively, compared to the mean for random chance of 49% (35% and 65% confidence interval).

## 0.10 Genetically identical barcodes are geographically clustered

**0.10.1 Population level comparisons.** Genetically identical barcodes were spatially clustered in Chad. Identical barcodes were within a median 18.6 kilometer range (standard deviation = 82.5 kilometers), and often within an approximately 50 kilometer radius (Fig 2A).

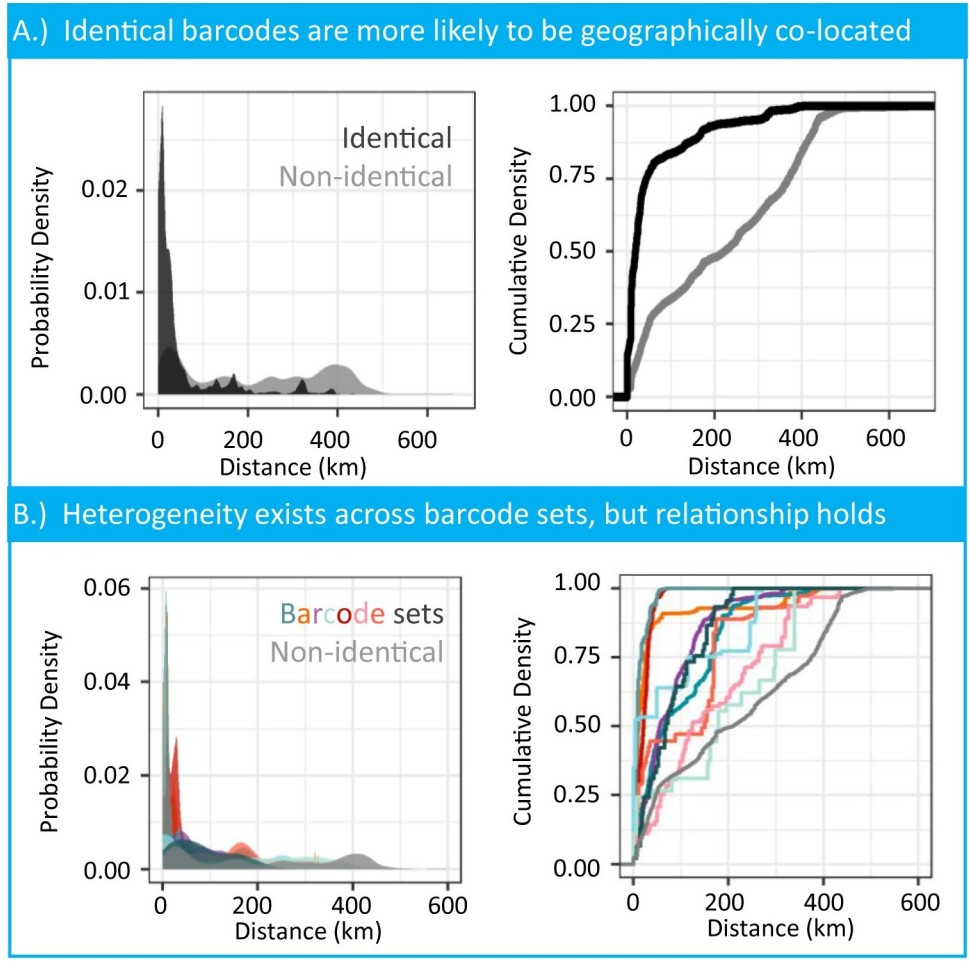

**Fig 2. Distance density estimates of pairwise similarity for identical and non-identical barcodes.** A.) Pairwise comparisons of identical barcodes in black (n = 10, 695) to a null of non-identical barcodes in gray (n = 79, 830). This distribution at 400 kilometers is the max distance between sampled cases along the Chari River. This enrichment of pairwise sample distance around 400 kilometers should not be considered the transmission upper bound. B.) The number of comparisons for each barcode are the unique pairs of all worms sharing that barcode identifier. The null distribution in gray of non-identical barcodes is subset to only include comparisons where one of the common barcodes must be found in the pair (n = 56, 913).

Non-identical barcodes were more evenly distributed within a median 222.2 kilometers (standard deviation = 157.5 kilometers, Fig 2A). The spatial distributions between identical and non-identical barcode distributions were statistically different (Kolmogorov–Smirnov test, p < 0.001). Reclassifying pairs of worms with a single variant difference to account for the chance of underestimated mutation rates or sequencing errors does not alter our findings (S2 Fig). Permuting the distance between pairs of worms removed spatial clustering for identical worms (S3 Fig).

Geographic clustering of barcodes was not due to the similarity of parasites within hosts that carry multiple infections. Sixty-five of the 245 hosts had multiple worm infections, with a median of 4 worms (range = 2–24) per multi-infected host (S4 Fig). Worms captured from the same host were not always genetically identical. The distribution of genetic similarity for worms from the same host did not differ from worms with the same GPS coordinates from

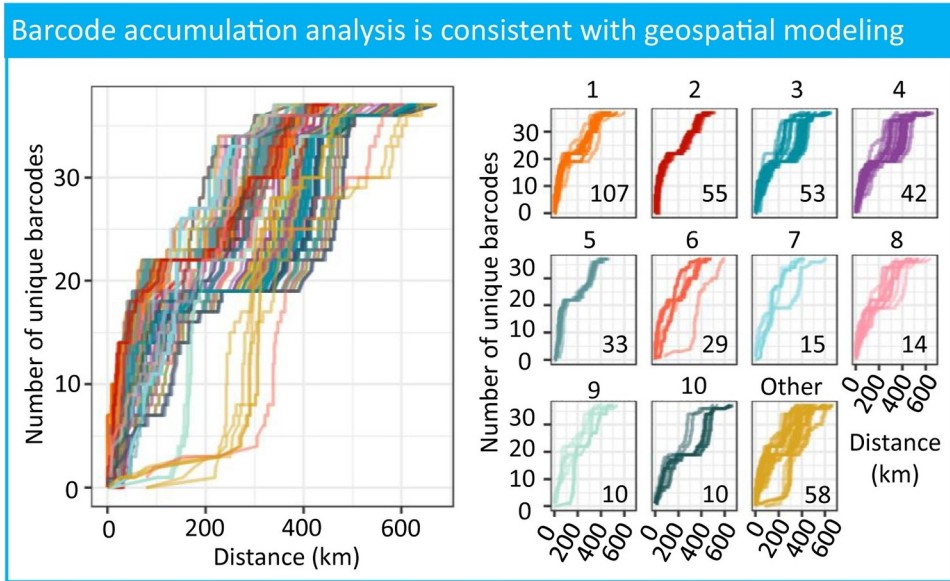

**Fig 3. Opportunistic sampling does not drive apparent spatial connectedness of samples.** Accumulation curves of worm cases on geographical proximity of barcode diversity show a consistent addition of diversity in quantitatively identified transmission ranges of barcodes sets (<50 kilometers). The left figure shows all accumulation curves for each worm case, and the right figure is separated by the barcode set number (1–10 for common barcodes, other for barcodes found in less than 10 samples). The numbers in the lower right corner are the number of worms in each barcode set.

different hosts (S4 Fig). Bootstrap subsampling of a single worm from each host maintained spatial clustering with low variability (S5 Fig).

**0.10.2 Individual barcode comparisons.** Individual barcodes showed variation in their geographical range (Fig 2B). For example in Fig 1C, cases with barcode 2 were found in the southeast mouth of the Chari River, while cases with barcode 3 and 4 were spread along the entire river. The geographic range for samples in each common barcode can be seen in S6 Fig. We replicated the smooth kernel density estimates for 10 of the 38 barcodes; note that three of the barcodes do not contain associated GPS locations for any sample. The geographic variability was not correlated to the total number of worms assigned to a barcode (S2 Table). Two of the barcodes seen in Fig 2B represented with pink and mint were more similar to the null distribution than other barcodes, driven by their broader geographic range in Chad.

The variation in geographical ranges for barcode groups was not driven by geographical isolation. For most worms, we observed a steep increase in unique barcodes up to 200 kilometers (Fig 3). Worms with a slow increase in the unique barcodes after 200 kilometers corresponded to the human cases observed furthest east (Fig 1A). These worms followed a similar steep increase once the radius includes samples along the Chari River. The plateau of diversity observed in most worms around 18–20 unique barcodes is an artifact of opportunistic sampling proximity (S7 Fig).

## 0.11 Expanding genetic markers can improve sensitivity for comparing worm populations

The extraction protocol of Guinea worm from an infected host results in degraded parasite DNA and a microbial mixture that is problematic for standard extraction and untargeted sequencing protocols. This limitation has dampened large scale genetic studies of Guinea

worm using additional markers in the mitochondrial and nuclear genome. It was unclear whether the untargeted regions of the mitochondrial genome would significantly alter population-level worm relatedness enough to consider further protocol development for complete characterization. To determine the value of including variants outside of the genes targeted in this study, we compared variants in other regions of the mitochondrial genome from an independent set of 19 worm samples with successful DNA extraction and shotgun sequencing [12].

Including mitochondrial variants outside of CO3, cytB, and ND3–5 gene ranges changed the distribution of genetic similarity between pairs of worms (Fig 4). Forty-seven variants were found within genes targeted in this study, and 129 variants within untargeted genes. Variants within targeted genes differentiated two peaks in the density estimates compared to all variants and variants within untargeted genes that differentiated three peaks in the density estimates. The distribution of genetic similarity between worms using the 47 targeted gene variants were statistically different than the distribution of genetic similarity using the 129 untargeted gene variants (Kolmogorov–Smirnov test, p < 0.001). The broader pairwise similarity distribution of untargeted genes was robust to the number of variants in the pairwise calculation (Fig 4). Distributions of randomly subsampled 47 untargeted gene variants compared to the distribution of 47 targeted gene variants showed that in some instances the population was less genetically identical with major peaks shifted to the left, and maintaining a higher density in the 0.75–0.85 pairwise genetic similarity range (Fig 4). However, because of the small sample size without accompanying GPS coordinates of this independent population, we could not extrapolate the effect of barcode sets on the spatial links between identical and non-identical pairs.

Despite requiring a minimum of 10 reads for each variant at a position to ensure true variant signals, we additionally confirmed the change in pairwise genetic similarity excluding

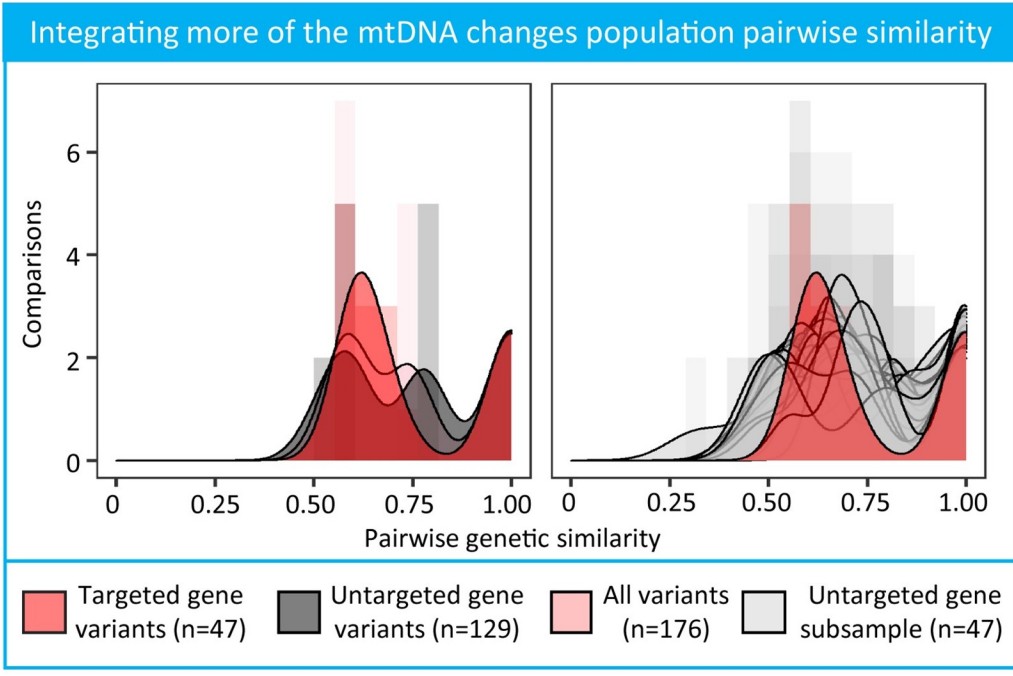

**Fig 4. Distribution of all-by-all pairwise genetic similarity of variants within or outside of amplification targeted genes.** Targeted gene variant regions were determined with CO3, cytB, and ND3–5 genomic ranges specified in the mitochondrial reference genome. Smoothed density kernel distributions are overlayed on pairwise genetic similarity distribution histograms for the different genomic regions.

variants observed in one sample in the population. Excluding variants found in only one worm, we dropped from 176 variants to 135 variants for comparison across the mitochondrial genome. The shape of the pairwise distribution including variants identified outside of the CO3, cytB, and ND3–5 gene ranges had three mass peaks compared to two mass peaks when only considering variants within the CO3, cytB, and ND3–5 gene ranges (S8 Fig), Kolmogorov–Smirnov test p = 0.001). The robust trend of a broader distribution when including singleton variants showed the pairwise distance between pairs of worms was not driven by unique variants in the population. Variants were called with an orthogonal method to circumvent any issues with genotype calling on a small sample size in GATK. More variants across the mitochondrial genome were identified calling with bcftools than GATK (185 versus 176 variants). A single variant from the GATK set was not represented in the bcftools set; 137 variants overlapped in reference and alternative alleles for each variant position between the two variant calling methods. For the remaining 38 variants, bcftools had one less alternative allele call compared to GATK tools. Despite these differences, variants called with bcftools replicated the different distributions for variants within targeted genes and outside targeted genes with a new mass peak around 0.75 pairwise genetic similarity. (Kolmogorov–Smirnov test, p = 0.001, S9 Fig).

## Discussion

To our knowledge, this is the largest retrospective study to date on Guinea worm transmission in Chad that links surveillance and genetic data with samples from 2012–2018, excluding 2014. Knowing the geographical range of transmission for genetically linked samples has the potential to be an important monitoring and evaluation tool for the elimination campaign. We identified 41 unique molecular barcodes from the 459 Chadian worm samples, with 426 samples having associated GPS data. The analyses in this study provide quantitative boundaries on the geographic range of transmission, with the majority occurring within a median 18.6 kilometer radius, and often within an approximately 50 kilometer radius. These results suggest the most effective interventions should consider case sweeps, water monitoring, and abatement approximately 20 kilometers around a reported case to reduce the spread of genetically related parasites. The population distribution falling under 50 kilometers implies that extending these efforts from 20–50 kilometers would largely dampen the dispersion of genetically related worms, with diminishing returns for efforts greater than 50 kilometers from a reported case. However, worms linked by barcode identity are not geographically isolated populations. Samples are surrounded by worms representing five to ten different barcodes within a 50 kilometer radius (Fig 3), and individual barcodes can vary in their transmission range from three to 150 kilometers. Geographically imposed interventions from these results need to consider the dispersion range variation for cases identified in areas with co-occurring lineages.

The modeling results presented in this article are consistent with previous analyses, but also expand the scope of geographic and genetic relatedness. Overlapping barcodes of worms collected from human and dog hosts supports the earlier conclusion by Thiele et al. [10] that humans and dogs share a similar Guinea worm population in their analysis of 75 samples (S1 (A) Fig). Barcodes did not consistently shift from one host species over time, suggesting a fluid transmission of *D. medinensis* between humans and dogs. Previous genetic analyses using spatial principal components analysis had identified a geographic trend of genetic relatedness down the Chari River in Chad [10, 38, 45]. The research in this article broadens that analysis by revealing that genetically identical and near-identical samples cluster geographically for multiple areas across Chad. (Fig 1C).

The findings of this work also align with the known epidemiology of the disease and biology of the parasite. Recent surveillance efforts involving the collaring and GPS tracking of dogs in Chad showed dogs visit water sources within a 10 kilometer range with variation across study sites [54]. The substantial geographic distance distribution for genetically identical pairs of worms under 50 kilometers are broadly in agreement with the dog roaming range and variation by geography (Fig 2). Both of these research efforts support the current epidemiological intuition and hypotheses about geographic connectivity of infections and the role of dogs as a reservoir [10, 46, 54].

The numerous sets of identical or nearly identical molecular barcodes among the 459 samples was expected. We anticipated some barcode sets would be maintained throughout the few years represented in this study due to the parasite biology of a year-long life cycle and the suspected low spontaneous mutation rate (see section 0.8). Even though the probability of having observed a direct transmission event in the available genetic samples is quite low (given the total number of reported cases), the molecular barcodes demonstrate distinct value even though the transmission network is only partially observed. The direct maternal inheritance of mitochondrial DNA and the persistence of barcodes across available years suggests a sustained population of worms are related through transmission. Taken together, these findings strongly suggest an epidemiological connection identified using the genetic data and can help inform the local epidemiology of Guinea worm in Chad.

There are several limitations to the analysis and modeling in this article. The samples that were sequenced and included in this analysis were retrospectively selected in order to span the geography of Chad and not through a systematic sampling frame. As mentioned, this led to an absence of 2014 samples in this study. We evaluated the robustness of each conclusion with sensitivity analyses to address the challenges posed by data constraints and methodologies (Figs 1 and 2). We demonstrated the geographic clustering results were not sensitive to dogs with multiple worm infections which could have artificially inflated the geographic proximity between pairs of worms (S5 Fig).

In addition, the characteristics of the *Dracunculus medinensis* genome are not as well-understood as viruses, bacteria, or other parasites. Without knowledge of baseline genetic diversity or a mutation rate for Guinea worm, we could not more accurately assess the genetic similarity threshold to group lineages. Despite these constraints, we showed that allowing for a single base pair difference in barcodes between a pair of worms maintained spatial clustering relative to two or more base pair differences in barcodes between a pair of worms (S2 Fig). Lineage representation is likely limited by the use of four mitochondrial genes coupled with the small sample size relative to the number of observed and unobserved total cases from 2015–2017. In an independent population of samples with complete coverage of the mitochondrial genome, we confirmed variants in other regions of the mitochondrial genome changed the distribution of population pairwise genetic relatedness (see section 0.11). This trend was replicated with a combination of bootstrapping and variant callers. Due to small sample sizes of available whole mitochondrial DNA, we were unable to conclude whether the extra variants refine geographic clustering. The substantial number of additional variants in the mitochondrial genome strongly suggests that access to more of the genome will better resolve identical lineages for Guinea worm.

Despite these limitations, the modeling and analyses in this article have important implications for policy makers and elimination programs. From 2010–2019, there was a concurrent increase in both reported cases and surveillance efforts. A complete characterization of the genetic diversity could help distinguish whether Guinea worm prevalence was increasing or a consequence of improved surveillance. The continued appearance of genetically identical worms across years suggests genomic data is informative for understanding transmission,

surveillance, and even intervention efficacy. Monitoring the genetic landscape could provide programmatic evidence for the effectiveness of geographically localized interventions by observing the potential elimination of barcode lineages. Sustained barcodes are particularly useful in instances where case reports may be disrupted due to insecurity or inaccessibility.

A comprehensive bank of all genomic samples paired with geographic data would allow a broader set of analyses to help the elimination program, for example use in outbreak analysis, characterization of importation versus local circulation, or to reveal potentially unknown animal reservoirs. Currently, we cannot provide programmatic guidance on a specific number of samples that should be collected and sequenced to capture the genetic diversity in Chad due to the high diversity of the *currently* sampled population (S1 Appendix). However, if a subset of historical samples share barcodes with the currently sampled population we would likely be powered to define a sample size for genetic surveillance. Conversely, more unique barcodes in historical samples would support the need to sequence all available worm samples due to the high diversity of the population. Current surveillance protocols already collect emerging worms in addition to a standardized survey for all reported cases. Given the reported case counts and access to high-throughput sequencing technologies, it is tractable to sequence a large subset or all of the available retrospective and prospective Guinea worm samples.

The confirmed change in population pairwise genetic similarity across the whole mitochondrial genome suggests expanding the marker set is an important future research direction. An expanded marker set would complement innovations constructing phylogenies from whole genome sequencing of the Guinea worm larvae [55] and microsatellites of the worm nuclear genome [10]. A holistic program of sequencing strategies and analytic methodologies will help translate research insights into programmatic input for the elimination of Guinea worm in Chad. Furthermore, additional analyses of the full range of epidemiological data collected by the CGWEP alongside linked genomic data are warranted, and may further elucidate transmission dynamics in Chad. The parasite genome has the potential to be an integral tool for the end-game strategy in Chad and beyond.

## Supporting information

**S1 Appendix. Predicting the increase in barcode variety with additional sequencing.** (PDF)

**S1 Table. Sequenced worms metadata.** Collection year, host, GPS coordinate availability, and whether included in Thiele et al. 2018 [10] for each sample. (CSV)

**S2 Table. Barcode counts for sequenced samples.** GPS columns indicate the number of samples for each barcode used in geospatial analyses by species. Columns starting with 'All' indicate the number of worms assigned to each barcode, including samples that were not linked to GPS data for each host species. Temporal presence/absence of barcodes can be tracked in S1(B) Fig per species. (CSV)

**S1 Fig. Molecular barcode comparison of worm barcodes identified per species.** A. The presence of different barcodes per host by year. A blue-filled cell next to each barcode is indicative that the barcode was represented at least once in the respective year. Refer to S2 Table for sample counts of each barcode by species. B. DAPC eigenvalues for principal components with all 41 unique barcodes. A drop in variance is observed from 4 to 5 components, but cross-validation suggests 15 components provide the highest classification success. C. Root mean square error (RMSE) ranges for components with different training set percentages. While

using 90% of the data as the training set produces the highest RMSE distribution, it obtains the lowest RMSE when compared to the 70% or 80% training set scenarios. D. DAPC analyses by number of components. Barcodes found in samples from humans only were grouped with barcodes found in samples from humans and dogs to create more even groups with barcodes found only in samples from dogs (humans only = 5 barcodes, both = 12, dogs only = 24).
(TIF)

**S2 Fig. Spatiotemporal modeling of barcode relatedness collapsed by a single base pair difference.** Number of pairwise comparisons for identical barcodes = 13,967, for non-identical barcodes = 76,558.
(TIF)

**S3 Fig. Permutation of distance between cases on barcode relatedness densities.** The x-axis is the permuted distance between pairs of worms and the y-axis represents the density for population genetic similarity scores for all worms (n = 426). Each line represents worm pair distance permutation (n = 100) for the population. The lines for identical and non-identical barcodes are consistent between permutations and overlap.
(TIF)

**S4 Fig. Barcode identity within hosts and genetic similarity distribution by shared location.** A. Each multi-infected host with the number of worms pertaining to each barcode set. "Not common" refers to barcodes found in less than 10 samples in the population for visual clarity. B. Distributions of genetic similarity between worms with the same reported GPS coordinates, colored by whether the pair is obtained from the same or different hosts.
(TIF)

**S5 Fig. Host subsampling on barcode relatedness densities.** The x-axis is the distance between pairs of worms and the y-axis represents the density for population genetic similarity scores with one worm per host (n = 282). Each line represents a bootstrap (n = 100) of a single worm per host.
(TIF)

**S6 Fig. Location of common and not common barcodes in Southern Chad.** Samples are colored by their barcode identifier, and shapes represent the host species. Barcodes 3, 4, and 10 have similar spatial distributions. Barcodes 2 and 5 have similar spatial distributions and are highly focal. Some barcodes span a very large geographic area, which suggests they are ancestral sequences that have diffused over time or are transmitting due to human behaviors.
(TIF)

**S7 Fig. Geographic distance between cases for each worm.** The geographic distance between worms relative to a single worm were organized in ascending order. The x-axis is the cumulative worm count by distance and the y-axis is the cumulative distance from each index worm. Spans of a flattened curve are indicative of geographic stretches that do not contain any samples and support the lack of barcode diversity observed in certain geographic distances in Fig 3.
(TIF)

**S8 Fig. Distribution of pairwise similarity using GATK variants found in greater than one worm.** The x-axis is the measured genetic similarity for variants within genes targeted by the amplification protocol (n = 35) and extending to the rest of the mitochondrial genome (n = 100), and y-axis is the number of pairwise comparisons (19*19 = 361). Filled regions

show the smoothed density estimates for histograms.
(TIF)

**S9 Fig. Orthogonal validation of population pairwise similarity using variants identified by bcftools mpileup.** The x-axis is the measured genetic similarity for variants within genes targeted by the amplification protocol (n = 47) and extending to the rest of the mitochondrial genome (n = 138), and y-axis is the number of pairwise comparisons (19*19 = 361). Filled regions show the smoothed density estimates for histograms.
(TIF)

## Acknowledgments

The authors would like to thank Dr. Fernando Torres for his helpful discussions regarding Guinea worm eradication efforts and Dr. Albert Lee for modeling discussions. This work would not be possible without the field and surveillance teams of the CGWEP. Thank you for the efforts of all involved to collect the epidemiological data and samples. The findings and conclusions in this report are those of the author(s) and do not necessarily represent the official position of the Centers for Disease Control and Prevention/the Agency for Toxic Substances and Disease Registry.

## Author Contributions

**Conceptualization:** Jessica V. Ribado, Guillaume Chabot-Couture, Joshua L. Proctor.

**Data curation:** Jessica V. Ribado, Nancy J. Li, Elizabeth Thiele, Sarah Anne J. Guagliardo.

**Formal analysis:** Jessica V. Ribado, Nancy J. Li, Hil Lyons.

**Funding acquisition:** Elizabeth Thiele.

**Methodology:** Jessica V. Ribado, Elizabeth Thiele, Hil Lyons, Joshua L. Proctor.

**Project administration:** Adam Weiss, Philippe Tchindebet Ouakou, Tchonfienet Moundai, Hubert Zirimwabagabo.

**Resources:** Elizabeth Thiele.

**Supervision:** Guillaume Chabot-Couture, Joshua L. Proctor.

**Visualization:** Jessica V. Ribado.

**Writing – original draft:** Jessica V. Ribado, Hil Lyons, Joshua L. Proctor.

**Writing – review & editing:** Jessica V. Ribado, Elizabeth Thiele, Hil Lyons, James A. Cotton, Sarah Anne J. Guagliardo, Guillaume Chabot-Couture, Joshua L. Proctor.

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
