## [Decision Letter · Decision Letter 0]

5 Dec 2020

Dear Dr. Ribado,

Thank you very much for submitting your manuscript "Linked surveillance and genetic data uncovers programmatically relevant geographic scale of Guinea worm transmission in Chad" for consideration at PLOS Neglected Tropical Diseases. As with all papers reviewed by the journal, your manuscript was reviewed by members of the editorial board and by several independent reviewers. In light of the reviews (below this email), we would like to invite the resubmission of a significantly-revised version that takes into account the reviewers' comments. 

All reviewers expressed positive opinions about the paper and its significance. Despite this, the reviewers all felt that improvement of the MS was possible, and they have provided useful guidance to assist this.

Reviewers emphasized a few points that I also noticed and that need to be addressed. First among these is the point about “singletons” and how they were treated. I found this very hard to follow but understanding what you mean is important for comprehension of later analyses. A simple, brief, explanation is likely possible and I hope to see that in a revised version. 

The mitochondrial genome in animals is not subject to recombination: it is effectively a single locus. Different parts of the mt genome may accumulate mutations at different rates, but all should tell the same story, subject to the range of error. So, I was confused at first about use of the word “locus” and the idea that some variants were apparently outside loci and others inside. It became clear that you were referring to a protein-coding gene as a “locus” and therefore regarded variants in non-coding regions as “outside” the locus. I found this usage confusing. It would be better to replace “locus” with “protein-coding gene” or similar.

Other small points.

Line 7. I don’t think that “dracunculiasis” should start with an upper-case letter.

Line 10. Hyphenate “decade-long”.

Line 79. “Alleles”? I would have called them “variants”, as elsewhere.

Line 232.”…30 human hosts…”

Line 233. “Of” at the end of the line seems unnecessary.

Line 266. What is meant by “Only four barcodes from this set (7, 21, 40 and 41) were observed as a singular event.” Maybe “only four barcodes were each found in a single worm”? In places, the language is overblown: this is one example.

Line 423. Something is wrong here. The word “between” should maybe be deleted.

Caption to Fig. 2 has a couple of typos.

We cannot make any decision about publication until we have seen the revised manuscript and your response to the reviewers' comments. Your revised manuscript is also likely to be sent to reviewers for further evaluation.

Sincerely,

David Blair

Associate Editor

Banchob Sripa

Deputy Editor

The reviewers emphasized a few points that I also noticed and that need to be addressed. First among these is the point about “singletons” and how they were treated. I found this very hard to follow but understanding what you mean is important for comprehension of later analyses. A simple, brief, explanation is likely possible and I hope to see that in a revised version. 

The mitochondrial genome in animals is not subject to recombination: it is effectively a single locus. Different parts of the mt genome may accumulate mutations at different rates, but all should tell the same story, subject to the range of error. So, I was confused at first about use of the word “locus” and the idea that some variants were apparently outside loci and others inside. It became clear that you were referring to a protein-coding gene as a “locus” and therefore regarded variants in non-coding regions as “outside” the locus. I found this usage confusing. It would be better to replace “locus” with “protein-coding gene” or similar.

Other small points.

Line 7. I don’t think that “dracunculiasis” should start with an upper-case letter.

Line 10. Hyphenate “decade-long”.

Line 79. “Alleles”? I would have called them “variants”, as elsewhere.

Line 232.”…30 human hosts…”

Line 233. “Of” at the end of the line seems unnecessary.

Line 266. What is meant by “Only four barcodes from this set (7, 21, 40 and 41) were observed as a singular event.” Maybe “only four barcodes were each found in a single worm”? In places, the language is over-complicated: this is one example.

Line 423. Something is wrong here. The word “between” should maybe be deleted.

Caption to Fig. 2 has a couple of typos.

Reviewer's Responses to Questions

**Key Review Criteria Required for Acceptance?**

**Methods**

-Are the objectives of the study clearly articulated with a clear testable hypothesis stated?

-Is the study design appropriate to address the stated objectives?

-Is the population clearly described and appropriate for the hypothesis being tested?

-Is the sample size sufficient to ensure adequate power to address the hypothesis being tested?

-Were correct statistical analysis used to support conclusions?

-Are there concerns about ethical or regulatory requirements being met?

Reviewer #1: The objectives of the study are articulated; the study design is appropriate; the population is clearly described with a sample size that is excellent for a population genetic study.

(1) Section 0.2.1 The description of how singletons were handled was a bit confusing. If I understand correctly, two worms both had “inflated” singletons (how many singletons count as inflated?), which suggests sequencing errors, and were excluded. For the remaining worms, six sites contained singletons and were removed from downstream analyses as they are not informative about relatedness between individual worms.

(2) Section 0.2.2 The use of the term “untargeted” can be removed as it's confusing—the entire mitochondrial genome was used, not just the portions that were not amplified and sequenced, so just saying “complete mitochondrial genome” is sufficient. Were optical duplicate reads and reads that mapped to more than one location in the genome removed in the analysis? If not, then this would need to be done; if it is part of the pipeline, please state this explicitly.

(3) Section 0.3 In the phylogenetic and population genetic research communities, UPGMA has been regarded for decades as a poor estimator of phylogenetic relatedness. Other methods are expected to be more accurate if the input data are sufficiently informative (maximum likelihood, Bayesian approaches, or even neighbor joining). For their phylogenetic analysis, the authors assume the simplest model of sequence evolution. This is an assumption that needs to be tested prior to running the analysis. There is no attempt to measure statistical support for the tree; in a maximum likelihood framework, bootstrapping can provide as a commonly used metric for statistical support, while in a Bayesian framework, posterior probabilities indicate the strength of support for a particular relationship. Finally, changes over time cannot be explored on the basis of this tree because the tree is not rooted, which is an essential component to directional analysis. Phylogenetic methods are used in viral epidemiology because viruses evolve very quickly within a single host. This is not true for parasites. I suspect that a phylogenetic tree built using solid phylogenetic methods would end up being a polytomy (i.e., no strong support for relationships between haplotypes). As an alternative to a phylogenetic approach, I might recommend instead applying population genetic methods that examine differences/changes in allele frequencies (i.e., the proportion of haplotypes in a given year). For example, discriminant analysis of principal components (DAPC) has easy-to-follow tutorials available online (Jombart et al. 2010). This could also be applied to produce a lovely visual comparison of how human vs dog haplotypes cannot be discriminated. That said, my intent here is not to push one particular method, but rather, to encourage the authors to increase the sophistication of whatever method they do use so that their conclusions are robust and well-supported statistically.

(4) Section 0.5.2 The authors calculate an expected diversity in the sample based on the mutation rate/generation x the length of the mitochondrial genome. This is a mis-application of this kind of mutation rate—it is not a substitution rate. The mutations Denver et al. quantify occur during oogenesis/gamete development, and any expected diversity that could be calculated would be between a mother and her offspring. It can't be used simply to get an estimate of population diversity. The usefulness of the mutation rate estimate based on C. elegans for population genetics is that it can be used to parameterize coalescent models, which are extremely useful for a range of questions in population genetics and in some applications in phylogenetics.

(5) Section 0.5.3 If I understand correctly, f is an average uncorrected pairwise distance between communities. This is a metric that has been previously described in the population genetic literature and should be cited.

(6) Section 0.6 When comparing the diversity of COIII, cytb, and ND2-5 to other parts of the genome, there is no consideration of why variation may be higher in other regions of the genome based on genome annotation. The reason this is important is that some regions of the genome evolve differently than other regions. For example, regions that are non-coding (less of the mitochondrial genome than nuclear in animals) tend to have a higher diversity than coding regions. In contrast, some genes are expected to be under stronger selective constraints than others. Genes that code for rRNA/tRNA tend to evolve differently than genes that code for proteins. Finally, primers for PCR-based studies are designed to bind to conserved regions to ensure robust and repeatable PCR across diverse samples. Thus, comparing the diversity as has been done here is not informative about future marker/barcode development. Rather than re-run these analyses taking into account how evolution is expected to act differently on variation in different parts of the mitochondrial genome, another option is to simply remove these analyses from the paper entirely. The take-home message is that sequencing more of the genome results in more information to distinguish worms and that some regions of the mitochondrial genome might be more variable than others, neither of which is novel or unexpected, and the same point could be made in the discussion by citing existing literature.

Reviewer #2: I find the methods confusing and not easy to follow. References are lacking for many of the methods used. It is also not clear as to how the Sanger sequences were checked and edited to rule out errors. The terminology needs to be reviewed in places to articulate what is an analysis term versus a biological term. For example "worm pairs" is used a lot and this must relate to between sample analyses rather than pairs of worms ? 

No ethics are detailed and as several of the samples are from humans this needs including in the methods. Also, anything related to live animals.

Reviewer #3: Yes to all of the above except the last question. I see no ethical or regulatory issues.

**Results**

-Does the analysis presented match the analysis plan?

-Are the results clearly and completely presented?

-Are the figures (Tables, Images) of sufficient quality for clarity?

Reviewer #1: The results as presented are adequate.

Reviewer #2: I feel the results could be much clearer. There is some very good data but unless you have a modelling background then it is not easy to follow. The main outputs from this study will be of interest in terms of the biology of transmission and so the methods and results should be more tailored to that audience. 

Sequence data should also be made available on a database such as Genbank.

Reviewer #3: Figures, Supplementary Figures, and Supplementary Information:

Please see the marked up copy with specific suggestions. There are several typos within the legends and some suggestions (like the addition of scale bars to maps), that may improve the figures. Also, in the current formatting, I found the figures to be too small. For example, it was hard to see distribution of the dots in Figure 1a and 1c. In addition, the labels in S4 were hard to read.

**Conclusions**

-Are the conclusions supported by the data presented?

-Are the limitations of analysis clearly described?

-Do the authors discuss how these data can be helpful to advance our understanding of the topic under study?

-Is public health relevance addressed?

Reviewer #1: The authors make a convincing case based on their analyses about the utility of parasite genetic data to inform strategies for elimination.

Reviewer #2: The conclusions are discussed together with the limitations but better structure is needed for this particularly as the limitations seem to far outweigh the conclusions.

Reviewer #3: Yes to all of the above.

**Editorial and Data Presentation Modifications?**

Reviewer #1: (No Response)

Reviewer #2: See more detailed comments below

Reviewer #3: Abstract

1) Background, consider hyphenating “ten-year absence”.

2) Methodology, specify “dog and human hosts” given the mention of paratenic hosts in the previous sentence.

3) Methodology, rephrase the last sentence to improve quality. A suggestion is in the attached manuscript.

4) Methodology, replace “three loci on” with “three loci in”

5) Consider revising the last sentence to improve clarity. A suggestion is in the attached manuscript. Also, this part of the abstract makes the least sense as there is no reference to “worm pairs” in earlier parts of the abstract. More of a transition to this part of the study in the Abstract would improve clarity.

6) Conclusions/Significance, revising a sentence will improve clarity – see suggestion in the attached manuscript.

Introduction

1) Line 8, specific that these losses are due to Dracunculiasis

2) Line 10, hyphenate “decade-long”

3) Line 27, hyphenate “genome-wide”

4) Line 38, delete “directly”. Using genetics to track the movement of parasites is an indirect way of assessing movement. You are making inferences which is based on host sample size, parasite sample size, variability of loci and number of loci, etc.

5) Line 43, improve clarity by revising sentence.

Methods and Materials

1) Line 69-71, clarify what “samples overlapped” means – the individual worm was part of the previous study? Also, delete extra “.”

2) Line 77-78 add the unit “bp” behind each reported range as the current formatting initially caused confusion.

3) Line 82, was comparing singleton counts among sequences the only method of quality control? Given that Sanger sequencing was used, were chromatograms examined for the sequences that were excluded?

4) Can line 83 (“Singleton variants identified in these two samples were excluded”) be deleted if these two sample were excluded (see previous line)?

5) Line 84-85. Why was the position missing a base call? Was this a quality issue or a deletion or insertion?

6) Line 90, revise the subheading to help remind readers why this part is included in the study.

7) Line 92, revise this sentence. A suggestion is provided in the attached manuscript.

8) Line 149, it is not clear what constitutes a “data-driving, empirical form”.

9) Line 179, what does this symbol mean, or is this a requirement of the journal? Replace with “see section 0.5.4?” Do this in other places is applicable.

10) Line 203, clarify what “population worm cases” are. How does one go from 459 to 425 worms?

11) Line 217, add a unit here to clarify that this is bp. Also, is the reference to PlasmoDB correct? If so, more explanation is needed.

12) Line 238-252, the methods associated with the barcode accumulation curve analysis should be included in the Methods section rather than the Results section to foreshadow the result. 

13) Reorder the methods to follow the order of the Results section. Put 0.2.2 last.

Discussion

1) Line 357, expand on why it is important to recognize diversity – especially this type of genetic diversity “mtDNA) in the discussion.

2) Line 396, replace “lead” with “led”

3) Line 404, move “other” to in front of “parasites”.

4) As someone who works on non-model parasites, I was curious as to what constitutes “a tailored set of phylogenetic or phylodynamic methodologies”.

5) Line 423, What is “between resolution”. Consider revising as “better resolution of genetic similarity between Guinea worms in Chad. Why would this only improve the resolution in Chad? Wouldn’t it improve it analyses of genetic similarity in other countries too?

**Summary and General Comments**

Reviewer #1: In this paper, the authors use an impressive number of samples from Chad to explore the relationships of genetic variation to geographic variation with the goal of building a model that can be used to estimate the risk of recrudescence should apparent elimination of Guinea worm in humans be achieved. This is an admirable and excellent goal, and the introduction and discussion are well-written and make an excellent argument for genetic epidemiology as a valuable approach. The authors convincingly build a case that identifying correlations between genetic diversity and geographic distance has practical applications for elimination of parasites. This research is a good first step.

However, the paper lacks diversity in citations of other literature on population genetics of parasites, primarily relying on previously published work on Guinea worms, and this is reflected in some problematic aspects of their analysis. There are some analyses that would need careful attention before I would recommend publication of the manuscript as-is. In particular, the methods used to generate the phylogenetic tree are inadequate, and the data need to be re-analyzed or replaced with a population genetic approach for answering the same questions (i.e., regarding dog and human sample clustering and changes in genetic diversity over time); the exploration of genetic diversity in the mitochondrial genome as a whole will need additional analysis to add more depth, or could be removed entirely.

Reviewer #2: This is an important study that aims to investigate the transmission dynamics of Guinea Worm to support eradication efforts. There is a large dataset analysed providing important and novel data, however the paper lacks clarity and is hard to follow with the key points lost within the analyses. Details are shown below and I suggest the authors revise the paper to make it more clear to the non-modelling audience. 

Abstract – worm pairs ? do you mean samples or worm pairs. 

Line 16: re-phrase the line. Genetic sequencing is not a tool for control but supports control through the understanding of transmission. Also change “Genetic sequencing” to “genetic analysis”. Sequencing is just one of many methods. 

Methods

A section on sample information is needed. E.g. what were the hosts for the 712 worms ? Dates of collections etc. A supplementary table would be useful. 

Lines 66-71 – make it more clear about the data from the samples as it is currently confusing. Maybe add a table of what DNA regions are available for each region. Also, was the data from reference 10 used in the analysis or was new data obtained from the samples used both studies. If the data was used then accession numbers need to be included etc. Make this much more clear. Add more information about how the data was obtained and edited. It is in reference 10 but good also to include some basic information and not just the reference. Include the loci etc. In the introduction it should also be stated that you are building on the data already analysed in reference 10. 

Line 74-89 – A typical sanger sequencing approach that the data was generated by allows for sequence editing and checking using sequence chromatograms. Can you confirm that this was done for the sequence data prior to alignment and allow you to produce contigs for each sample which can then be analysed. The methods described are confusing e.g. what do the genomic ranges relate too ? why is there missing data ? missing data would be due to sequence error and not as a variant ? you would not normally see deletions within a species ? What do you mean by singletons and why are some discarded and some kept ? I am also assuming that your barcode gives you a genotype so you identified 41 genotypes within the 459 samples ? It might also be worth investigating the SNP’s in terms of synonymous versus non-synonymous mutations. Was there also clear differences in the frequency of the different genotypes. 

Lines 90 109 – so you need to explain that this is NGS data as this is annotated and aligned in a different way to the sanger data, particularly in the way that errors and quality is assessed. This data also needed to be included in some sort of sample table showing what data comes from what sample and what methods etc. It is unclear if the whole mt genome was used in the analyses or just the target genes. 

It is not clear why the barcode method is better than a full sequence analysis, with model selection, bootstrapping, AA changes etc. 

Line 131 – this GPS data should also be added to the sample master file so the full data is available. 

Line 139 – refer to a table that details to number of worms analysed per host and what the hosts were. 

Line 144 -explain what you mean by yearly transmission cycle. Is it to do with pre-patent periods or seasonality etc. 

Spatial Analysis – would the spatial analysis be based on water bodies that are connected to allow spread not just distance between water bodies that allow movement of hosts between water bodies. 

Line 169 – you refer to sequencing error here. Sequencing error should be very low for the Sanger data so it is confusing as to why more sequence editing was not done. Also, could a genetic clustering analysis (e.g. a haplotype network or similar helps with grouping population clusters which could then have been analysed spatially. Also was the clustering analysed in terms of host ? 

Line 194 – refrain from using the term worm pairs as this suggests male and female pairs. Use a different terminology. 

Sensitivity analyses – is this the correct term or should it be statistical analyses. 

What does distance vector relate to – is it the disease vector (copepod). Do try and help the readers that are not modellers. 

Paragraph 202 – are there references to these methods used. You also need to be more clear on how this is done and refrain from complexed terminology e.g. what is the index worm and are the population worm cases just samples ? 

Paragraph 214 – more references need to be included for bit of the analyses used. It is difficult to understand what data the variants outside of the loci is based on ? 

Line 232 – Figure 1A says number of sequences per host for each year but this should be number of samples analysed there will be more than one sequence for each sample? 

Line 234 – case coordinates means geographical location of the infection ? make this more clear. 

Line 241 – this Binomial analyses should be detailed in the methods ? 

Line 267 – the analyses are based on mitochondrial DNA so diversity will be high and increase with sample size as seen in the dogs. Based on the mt DNA you do not know if the worms are genetically identical, they are the same genotype based on maternal inheritance but you cannot tell if they are clones ? 

Line 278 – what is a sequencing mutation ? do you mean error ? 

Line 309 – this is the information that was needed in the methods. 

Figure S4 – add better scale to the A – is the lower limit 1 – also the last host has lots of worms – this kind of extreme should note some discussion. The Y axis numbers are also unclear. For B use yes and no instead of false and true as false tends to mean unreliable. 

Figure S6 – you have non common barcodes shown too so change the figure title. 

Line 326 – do you mean single variants – variants found in only one worm suggests you get variation within the worm ? 

Line 327 – I think you need to make it more clear what you mean by within and outside loci. 

Line 343 – you did not analyse 459 sequences but 459 samples ? 

Line 345- is quantitative bounds the correct term ? do you mean boundaries ? 

In your discussion on limitations why was a full phylogenetic analysis not performed on the data and it was just based on the mutation barcode – this could have provided resolution to the analyses. 

Also data should also be made available through a data base such as Genbank. 

I think the discussion could be tightened to show the clear findings better. There are clearly many limitations to the study and these are discussed but I do not find the discussion well structured. 

Generally the paper shows important findings and advances the needed analyses for understanding the animal reservoirs for or the zoonotic transmission of Guinea Worm but the paper is very hard to follow, with non-uniform terminology and also terms that mean several things e.g. could relate to the biology of the parasite or the analyses. The authors need to consider the audience and provide clearer methods and results.

Reviewer #3: This manuscript builds upon previous molecular-ecology approaches to better understand the distribution of genetic diversity of Dracunculiasis in Chad. The dataset is expanded by considering more loci and more samples in space and time. The results show that worms with the same mtDNA multi-locus barcode tend to be clustered within a relatively small geographic range (within 50 km). However, these same barcodes are not partitioned by year or by host suggesting some fluid transmission dynamics. These findings are the result of comprehensive analyses and complimented by a further exploration of what these patterns would be like if the entire genome was analyzed. Perhaps unsurprisingly, modelling with genomes suggested that more loci would give better resolution into the genetic relationships among samples and would improve the resolution that genetic data could provide for programmatic surveillance and decision-making.

This manuscript stands out for its comprehensive set of analyses to investigate the validity of its various results. Also, I enjoyed reading the Discussion section. I recommend publication pending minor revisions. I included a list of these comments, but also copies of the MS and the supplemental materials with my comments annotated throughout.

PLOS authors have the option to publish the peer review history of their article (what does this mean?). If published, this will include your full peer review and any attached files.

Reviewer #1: No

Reviewer #2: No

Reviewer #3: No
---

## [Editor Report · Decision Letter 1]

24 Mar 2021

Dear Dr. Ribado,

Thank you very much for submitting your manuscript "Linked surveillance and genetic data uncovers programmatically relevant geographic scale of Guinea worm transmission in Chad" for consideration at PLOS Neglected Tropical Diseases. 

I have read the responses to reviewers and the revised version of the paper. The reviewers have provided excellent comments and guidance. I wish to avoid unnecessary rounds of review, so I am sending this back to you now for more work. The reviewers complained of lack of clarity in explanations of methods. In my opinion, many of the changes made in this revision have done little to improve clarity and, in a few cases, might have introduced additional problems. This is likely to irritate the reviewers, something best avoided. I am not asking you to change any analyses, just to look at the writing and try to make it clearer. Resubmit it, and then it can go back to reviewers.

I have flagged this in the system as “major revision”, but that might be excessive.

Some specifics. But you need to read the entire paper for flow and consistency. It remains verbose and could benefit from skilled editing.

I’ll refer to line numbers in the document PNTD-D-20-01771_R1_trackedChanges.pdf. 

Abstract (no lines numbers for the abstract in the document). 

Wrong use of the word “paratenic”

Are you sure you mean “…genetically identical worm populations…”?

The sentence “we investigated the value of…” should be reviewed. And it is hardly surprising that the addition of a longer region of the mt genome will distinguish more lineages.

In the third line of the “conclusions/significance” section of the abstract, is “co-occurrence” better than “overlap”? I have trouble envisaging two worms overlapping.

Line 23. You talk about “phylogenetic methodologies…”. But the phylogeny previously in the paper has been removed. Do you want to retain this part of the introduction?

Line 66. Something is missing after “successfully”. Should it be “amplified”? “sequenced”?

Line 70. Maybe “reported in previous literature”?

Lines 75, 98 and elsewhere. The adjectival form “mitochondrial” would be better.

Senetence starting on line 77. Is it needed? The sentence that follows seems to provide necessary information.

Line 78. Phrasing. I would say “The regions sequenced were from sites 3,690 to 4,308 for CO3…etc.” See also around lines 250-253.

Lines 84-93. Here, the attempt to clarify the way in which singletons were dealt with has not made things much clearer. Please revisit this section. I quite like “inflated singleton counts”, which you have removed in this revision. The next few points also relate to this section. 

Line 86. The sentence starting “Non-singleton variants were retained if…” confuses me. If a base (variant?) is present in "all but one sample", then surely that one sample represents a singleton?

Line 87. The sentence starting “Four samples had an other position…” does not make much sense.

Line 89. “These criteria”. Given the new material added to the paragraph, the reader might have trouble working out what criteria these are.

Line 92. Remove “see”.

Line 95. Surely it is not Dracunculus (note spelling and lack of italics) samples that were publicly available, but rather Dracunculus mitochondrial genomes?

Line 126. Rather than “of the molecular barcodes” I would say “based on the molecular…”

Line 131. Should the word not be “discriminant”?

Line 159. This sentence needs attention. I don’t think you can refer to a “pre-patent cycle”. Maybe something like “Here, given the annual life-cycle of guinea worm, …” 

Line 164. “…is a the…”?

Line 170 and elsewhere. How often do you need to say “the functional form of f”? I found the phrase at least three times. And should “f” be in italics or not? Be consistent.

Lines 175-186. Present and past tenses both used. Reporting what you did, the past tense is better. This switching between tenses is throughout the Methods and Results sections. Please check and be consistent. Results should also be in the past tense: reporting what you actually did.

Line 205. The sentence starting “Note” is hard to follow.

Line 214. “criteria” is a plural form.

Line 222 “worms pairs”?

Line 238.”The number … was counted…”

Lines 247-256. Revisit and try to increase brevity and clarity.

Lines 2720273. “…for each reported case”?

Line 273. The map in Fig. 1 does not identify the Chari River. 

Line 360. I would replace “with” with “using”.

Lines 399-403. Some of this seems to be stated also around lines 377-379. Check for repetition.

Line 413. Maybe “Knowing the geographical range of transmission”?

Line 436. Maybe “…and this suggests…”?

Line 516. “Sequencing more historical samples could either decrease genetic diversity…”. I don’t follow this.

Lines 521-524. This is an example of a very wordy sentence. Try to edit it (and others) to a more digestible form.

Fig 2B, caption. Would better phrasing be “For each barcode, pairwise distances among all pairs of worms with that barcode were calculated“?

Fig. 3, caption. Maybe “… does not drive apparent spatial connectedness”?

Your revised manuscript is also likely to be sent to reviewers for further evaluation.

Sincerely,

David Blair

Associate Editor

Banchob Sripa

Deputy Editor

I have read the responses to reviewers and the revised version of the paper. The reviewers have provided excellent comments and guidance. I wish to avoid unnecessary rounds of review, so I am sending this back to you now for more work. The reviewers complained of lack of clarity in explanations of methods. In my opinion, many of the changes made in this revision have done little to improve clarity and, in a few cases, might have introduced additional problems. This is likely to irritate the reviewers, something best avoided. I am not asking you to change any analyses, just to look at the writing and try to make it clearer. Resubmit it, and then it can go back to reviewers.

I have flagged this in the system as “major revision”, but that might be excessive.

Some specifics. But you need to read the entire paper for flow and consistency. It remains verbose and could benefit from skilled editing.

I’ll refer to line numbers in the document PNTD-D-20-01771_R1_trackedChanges.pdf. 

Abstract (no lines numbers for the abstract in the document). 

Wrong use of the word “paratenic”

Are you sure you mean “…genetically identical worm populations…”?

The sentence “we investigated the value of…” should be reviewed. And it is hardly surprising that the addition of a longer region of the mt genome will distinguish more lineages.

In the third line of the “conclusions/significance” section of the abstract, is “co-occurrence” better than “overlap”? I have trouble envisaging two worms overlapping.

Line 23. You talk about “phylogenetic methodologies…”. But the phylogeny previously in the paper has been removed. Do you want to retain this part of the introduction?

Line 66. Something is missing after “successfully”. Should it be “amplified”? “sequenced”?

Line 70. Maybe “reported in previous literature”?

Lines 75, 98 and elsewhere. The adjectival form “mitochondrial” would be better.

Senetence starting on line 77. Is it needed? The sentence that follows seems to provide necessary information.

Line 78. Phrasing. I would say “The regions sequenced were from sites 3,690 to 4,308 for CO3…etc.” See also around lines 250-253.

Lines 84-93. Here, the attempt to clarify the way in which singletons were dealt with has not made things much clearer. Please revisit this section. I quite like “inflated singleton counts”, which you have removed in this revision. The next few points also relate to this section. 

Line 86. The sentence starting “Non-singleton variants were retained if…” confuses me. If a base (variant?) is present in "all but one sample", then surely that one sample represents a singleton?

Line 87. The sentence starting “Four samples had an other position…” does not make much sense.

Line 89. “These criteria”. Given the new material added to the paragraph, the reader might have trouble working out what criteria these are.

Line 92. Remove “see”.

Line 95. Surely it is not Dracunculus (note spelling and lack of italics) samples that were publicly available, but rather Dracunculus mitochondrial genomes?

Line 126. Rather than “of the molecular barcodes” I would say “based on the molecular…”

Line 131. Should the word not be “discriminant”?

Line 159. This sentence needs attention. I don’t think you can refer to a “pre-patent cycle”. Maybe something like “Here, given the annual life-cycle of guinea worm, …” 

Line 164. “…is a the…”?

Line 170 and elsewhere. How often do you need to say “the functional form of f”? I found the phrase at least three times. And should “f” be in italics or not? Be consistent.

Lines 175-186. Present and past tenses both used. Reporting what you did, the past tense is better. This switching between tenses is throughout the Methods and Results sections. Please check and be consistent. Results should also be in the past tense: reporting what you actually did.

Line 205. The sentence starting “Note” is hard to follow.

Line 214. “criteria” is a plural form.

Line 222 “worms pairs”?

Line 238.”The number … was counted…”

Lines 247-256. Revisit and try to increase brevity and clarity.

Lines 2720273. “…for each reported case”?

Line 273. The map in Fig. 1 does not identify the Chari River. 

Line 360. I would replace “with” with “using”.

Lines 399-403. Some of this seems to be stated also around lines 377-379. Check for repetition.

Line 413. Maybe “Knowing the geographical range of transmission”?

Line 436. Maybe “…and this suggests…”?

Line 516. “Sequencing more historical samples could either decrease genetic diversity…”. I don’t follow this.

Lines 521-524. This is an example of a very wordy sentence. Try to edit it (and others) to a more digestible form.

Fig 2B, caption. Would better phrasing be “For each barcode, pairwise distances among all pairs of worms with that barcode were calculated“?

Fig. 3, caption. Maybe “… does not drive apparent spatial connectedness”?
---

## [Decision Letter · Decision Letter 2]

25 May 2021

Dear Dr. Ribado,

Thank you very much for submitting your manuscript "Linked surveillance and genetic data uncovers programmatically relevant geographic scale of Guinea worm transmission in Chad" for consideration at PLOS Neglected Tropical Diseases. As with all papers reviewed by the journal, your manuscript was reviewed by members of the editorial board and by several independent reviewers. The reviewers appreciated the attention to an important topic. Based on the reviews, we are likely to accept this manuscript for publication, providing that you modify the manuscript according to the review recommendations. 

The paper has greatly improved, but the reviewers still noted some points they wished to see improved. The English could be better in many places. Note also comments from reviewer #3 about use of the term “loci”. 

A few more details that I noticed:

Line 105. This sentence states that “Trained program and ministry staff … are stored in ethanol…”.

Lines 150 and 154 “mitochondrial genome”.

Line 163. Can a mitochondrial genome exhibit heterozygosity?

Line 198. “… yearly prepatent cycle…”? Reconsider this phrasing. The life cycle is annual.

Various places. “barcode x” sometimes starts with a capital letter, sometimes not. Be consistent.

Line 364. Is 47 the correct number? Elsewhere 86 was stated.

Line 386. “… this robust trend was also observed…”.

Line 391. “Variant calls…”.

Sincerely,

David Blair

Associate Editor

Banchob Sripa

Deputy Editor

The paper has greatly improved, but the reviewers still noted some points they wished to see improved. The English could be better in many places. Note also comments from reviewer #3 about use of the term “loci”. 

A few more details that I noticed:

Line 105. This sentence states that “Trained program and ministry staff … are stored in ethanol…”.

Lines 150 and 154 “mitochondrial genome”.

Line 163. Can a mitochondrial genome exhibit heterozygosity?

Line 198. “… yearly prepatent cycle…”? Reconsider this phrasing. The life cycle is annual.

Various places. “barcode x” sometimes starts with a capital letter, sometimes not. Be consistent.

Line 364. Is 47 the correct number? Elsewhere 86 was stated.

Line 386. “… this robust trend was also observed…”.

Line 391. “Variant calls…”.

Reviewer's Responses to Questions

**Key Review Criteria Required for Acceptance?**

**Methods**

-Are the objectives of the study clearly articulated with a clear testable hypothesis stated?

-Is the study design appropriate to address the stated objectives?

-Is the population clearly described and appropriate for the hypothesis being tested?

-Is the sample size sufficient to ensure adequate power to address the hypothesis being tested?

-Were correct statistical analysis used to support conclusions?

-Are there concerns about ethical or regulatory requirements being met?

Reviewer #1: The new analyses and clarifications have significantly improved the manuscript.

Section 0.4. Typically, cross-validation ("xval") is used to determine the optimal number of principal components to use in a DAPC. The approach used here, testing several different values for the number of PCs used in the analysis and reporting the qualitative similarities across replicates, is okay, depending on the distribution of variation in those PCs. The PC eigenvalues can be plotted relatively easily (scree.pca=TRUE). If the PCA eigenvalues reach a plateau by 5 PCs, then the number used should be acceptable. If not, then cross-validation will need to be used to determine how many PCs to include in the analysis.

Section 0.3.2. It is fairly standard to take the intersection of variant calls between alternative calling programs. I don't want to suggest re-doing analyses at this point, but if the barcode variants reported in the main text are found in both pipelines, it would be sufficient to indicate this rather than reporting variant distributions (lines 390-393/Fig S9).

Reviewer #2: The methods need more information on the samples used and to make clear what data is used from published datasets and what is new data as part of this study.

Reviewer #3: The revision of PNTD-D-20-01771_R2 is improved from the previous version. I have no major concerns about the methods used in this manuscript.

**Results**

-Does the analysis presented match the analysis plan?

-Are the results clearly and completely presented?

-Are the figures (Tables, Images) of sufficient quality for clarity?

Reviewer #1: The results and figures are acceptable.

Section 0.11/Figure 4. I am still not convinced that comparing the variants identified within loci (n=47) to the variants outside of loci (n=129) is an appropriate comparison given the limited sampling. The analysis does not discriminate between improved differentiation based on increased information (= more data) and improved differentiation based on different processes acting on variants located in these different categories. From the perspective of marker development, the comparison on the basis of whether a variant falls within or outside of the barcode used in the broader study is not useful. Comparing the 47 variants used on the larger sample to variants inferred from the whole genome is sufficient to make the point that increasing the amount of sequence data increases the information content.

Panel B of Fig 4 could be moved to supplemental, as it is not referred to in the discussion.

Reviewer #2: (No Response)

Reviewer #3: The revision of PNTD-D-20-01771_R2 is improved from the previous version. I have no major concerns about the results in this manuscript.

**Conclusions**

-Are the conclusions supported by the data presented?

-Are the limitations of analysis clearly described?

-Do the authors discuss how these data can be helpful to advance our understanding of the topic under study?

-Is public health relevance addressed?

Reviewer #1: The conclusions are well-supported by the data and limitations are discussed appropriately. This manuscript demonstrates that parasite genetic data has utility for programs seeking to eliminate NTDs that remain a public health problem.

Reviewer #2: The authors have responded to the necessary revisions

Reviewer #3: The revision of PNTD-D-20-01771_R2 is improved from the previous version. I have no major concerns about the results in this manuscript.

**Editorial and Data Presentation Modifications?**

Reviewer #1: line 52: dracunculiasis should not be capitalized

line 451: insert period between (Fig 1 and 2). We demonstrated...

line 454: change "is" to "are" ...genome are not as...

Supplemental Fig. S1: the x-axis should read "Discriminant function 1" (because more than 1 PC was used in this discriminant function)

Reviewer #2: Minor english corrections are needed and a careful read is suggested.

Reviewer #3: The authors have made many changes from the original MS and provided detailed responses to the points raised by the editor and reviewers. I recommend minor revisions aimed at providing further clarity in the writing especially to how the word “loci” is used.

As the editor suggested, replacing “loci” with “genes” if referring to the 3 targeted genes in the mtDNA genome would provide clarity and more accurately reflect that mtDNA arises from a single locus. Then the term “genotype” appears to be used correctly (lines 11-12). Otherwise, “multilocus” would seem more appropriate in that particular sentence, but this is in fact NOT correct since mtDNA arises from a single locus.

Changing “locus/loci” to “gene/genes” would affect many places in the MS including: lines 11-12, 137, 162, 327-346, 451. 455, and 487. Also, the caption for Fig 4.

In addition, it would be mean changing the terminology regarding “within loci” and “outside of loci” for lines 327-346, 470. An alternative possibility is to use “among targeted genes” and “among non-targeted” genes?

Also, the authors did use “gene” in lines 150-155. So seems a good reason to stay consistent.

Other minor comments:

Lines 20-22. It may be more direct to refer to several genes or >1 gene rather than markers in this sentence.

Line 72: correct spelling of “phlyogenetic”

Line 85-88: The way this revised sentence reads to me is that people are directly studying pathogen movement to indirectly study human movement. I would suggest that the reason genetic data is indirect is that we estimate pathogen population genetics to make inferences about human and pathogen movement. Or in other words, genetic data are used to indirectly study the movements of pathogens. 

Line 104: should this refer to mtDNA genes?

Line 126: replace with “were successfully sequenced”

Line 129: consider replacing with “first reported in a previous assessment of genetic diversity of Chadian Guinea worms”.

Line 259: Insert “of”

Line 447: Replace “on” with “in”

Line 457: Add space before parentheses.

Line 466: “Non-loci” is a misnomer?

Figures and Tables:

1) Typo in Fig 1 – see comments in manuscript.

2) hyphenate “blue-filled” in Supp Fig 1.

3) Delete “appear” in Supp Fig S6. 

4) Clarify what “current loci” refers to in Supp Fig S8.

**Summary and General Comments**

Reviewer #1: Eradication of many neglected tropical diseases is facing challenges in some regions. Key to successful elimination is identifying mechanisms that underlie continued transmission despite control measures. In this paper, genetic sequence data from Guinea worms are used to confirm dogs as a significant zoonotic reservoir, and to quantify the geographic scale of transmission. Developing genetic barcodes for use in surveillance should be considered (and debated) by the NTD community as a tool for program managers to clarify when movement of people or animal reservoirs drives continued transmission. The research presented here contributes to this discussion, and I recommend its publication.

Reviewer #2: The authors have responded to the necessary revisions but there are a few minor revisions as suggested.

Reviewer #3: This study demonstrates that barcoding and mitochondrial genome analysis can be used to better understand the distribution of the genetic diversity of dracunculiasis in Chad.

PLOS authors have the option to publish the peer review history of their article (what does this mean?). If published, this will include your full peer review and any attached files.

Reviewer #1: No

Reviewer #2: No

Reviewer #3: No

Figure Files:

Data Requirements:

Reproducibility:

References

---

## [Editor Report · Decision Letter 3]

27 Jun 2021

Dear Dr. Ribado,

Thank you very much for submitting your manuscript "Linked surveillance and genetic data uncovers programmatically relevant geographic scale of Guinea worm transmission in Chad" for consideration at PLOS Neglected Tropical Diseases. As with all papers reviewed by the journal, your manuscript was reviewed by members of the editorial board and by several independent reviewers. The reviewers appreciated the attention to an important topic. Based on the reviews, we are likely to accept this manuscript for publication, providing that you modify the manuscript according to the review recommendations. 

This version is much better and almost ready to be accepted. You have revised it very carefully: thanks. I noticed that in line 403 (line 421 in the tracked-changes version), you have written "from the from the". Fix this small detail and resubmit and I'll recommend acceptance. Obviously, there is n need for this to go back to reviewers.

Sincerely,

David Blair

Associate Editor

Banchob Sripa

Deputy Editor

This version is much better and almost ready to be accepted. You have revised it very carefully: thanks. I noticed that in line 403 (line 421 in the tracked-changes version), you have written "from the from the". Fix this small detail and resubmit and I'll recommend acceptance. Obviously, there is n need for this to go back to reviewers.

Figure Files:

Data Requirements:

Reproducibility:

References

---

## [Editor Report · Decision Letter 4]

29 Jun 2021

Dear Dr. Ribado,

We are pleased to inform you that your manuscript 'Linked surveillance and genetic data uncovers programmatically relevant geographic scale of Guinea worm transmission in Chad' has been provisionally accepted for publication in PLOS Neglected Tropical Diseases.

Best regards,

David Blair

Associate Editor

Banchob Sripa

Deputy Editor

Thanks for fixing that problem. I recommend acceptance.

---

## [Editor Report · Acceptance letter]

21 Jul 2021

Dear Dr. Ribado,

We are delighted to inform you that your manuscript, "Linked surveillance and genetic data uncovers programmatically relevant geographic scale of Guinea worm transmission in Chad," has been formally accepted for publication in PLOS Neglected Tropical Diseases.

Best regards,

Shaden Kamhawi

co-Editor-in-Chief

Paul Brindley

co-Editor-in-Chief
